# Antimicrobial Activity of Sertraline on *Listeria monocytogenes*

**DOI:** 10.3390/ijms24054678

**Published:** 2023-02-28

**Authors:** Yidi Wang, Lili Li, Pingyao Cai, Rikke Heidemann Olsen, Shuai Peng, Hecheng Meng

**Affiliations:** 1School of Food Science and Engineering, South China University of Technology, Guangzhou 510641, China; 2Institute of Food Safety and Nutrition, Jinan University, Guangzhou 510632, China; 3Department of Veterinary and Animal Sciences, Faculty of Health and Medical Sciences, University of Copenhagen, 1870 Frederiksberg, Denmark; 4Hubei Provincial Engineering Laboratory for Clean Production and High Value Utilization of Bio-Based Textile Materials, Wuhan Textile University, Wuhan 430073, China; 5College of Life Science, South-Central Minzu University, Wuhan 430074, China

**Keywords:** *Listeria monocytogenes*, sertraline, antimicrobial activity, biofilm, virulence

## Abstract

We explored the antimicrobial activity of sertraline on *Listeria monocytogenes* and further investigated the effects of sertraline on biofilm formation and the virulence gene expression of *L. monocytogenes*. The minimum inhibitory concentration and minimum bactericidal concentration for sertraline against *L. monocytogenes* were in the range of 16–32 μg/mL and 64 μg/mL, respectively. Sertraline-dependent damage of the cell membrane and a decrease in intracellular ATP and pH_in_ in *L. monocytogenes* were observed. In addition, sertraline reduced the biofilm formation efficiency of the *L. monocytogenes* strains. Importantly, low concentrations (0.1 μg/mL and 1 μg/mL) of sertraline significantly down-regulated the expression levels of various *L. monocytogens* virulence genes (*prfA*, *actA*, *degU*, *flaA*, *sigB*, *ltrC* and *sufS*). These results collectively suggest a role of sertraline for the control of *L. monocytogenes* in the food industry.

## 1. Introduction

*Listeria monocytogenes* (*L. monocytogenes*) is a Gram-positive pathogen and the causative agent of listeriosis [1] in which the typical clinical symptoms include meningitis, septicemia and stillbirth with a mortality rate of up to 30%, especially in immunocompromised hosts [2]. *L. monocytogenes* inhabits a broad ecologic niche and can contaminate various food products and food processing environments [3]. As this bacterium is psychrophilic; resistant to desiccation, acid and heat; and tolerant to increased sub-lethal concentrations of disinfectants or resistant to lethal concentrations [4], it can persist in food processing environments and poses a challenge to the food production industry.

*L. monocytogenes* can adhere to food-processing surfaces and form a biofilm on these surfaces [5]. The biofilm formation ability is essential for *L. monocytogenes* to survive and persist in food processing environments [5]. Biofilm is composed of microbial cells and self-produced extracellular polymeric substances (EPS), including polysaccharides, nucleus acids, proteins and lipids [6]. The EPS forms unique three-dimensional (3D) spatial structures and provides mechanical stability to the biofilm [6]. Notably, biofilm is closely related to functional properties, such as decreasing the efficiency of cleaning treatments and providing in-biofilm located bacteria with high resistance to antimicrobial agents [6]; for these reasons, bacterial biofilm is a severe threat to food safety. Thus, to secure public health, it is important for the food industry to find effective approaches to control the biofilm formation of *L. monocytogenes*.

Sertraline, a selective serotonin reuptake inhibitor originally commercialized as an antidepressant drug, is reported to possess antimicrobial activity against a wide range of bacteria, such as *Salmonella* spp., *Shigella* spp., *Staphylococcus* spp., *Staphylococcus epidermidis*, *Streptococcus* spp., *Vibrio* spp., etc. [7,8]. Although the antimicrobial activity of sertraline has attracted attention, it is not fully elucidated how this compound performs its functions, and little is known about its effects on biofilm and virulence gene expression.

In this study, we examined the antimicrobial activity of sertraline against *L. monocytogenes* and further analyzed its effects on biofilm formation and the virulence gene expression of *L. monocytogenes* strains.

## 2. Results

### 2.1. Antimicrobial Activity of Sertraline

The minimum inhibitory concentration (MIC) and minimum bactericidal concentration (MBC) values of sertraline for the *L. monocytogenes* strains were 16–32 μg/mL and 64 μg/mL, respectively (Table 1). Interestingly, the treatment of 64 μg/mL sertraline sharply decreased the bacterial concentration of both *L. monocytogenes* strains, which reached 0 CFU/mL at 12 h (Figure 1), indicating the bactericidal effect of sertraline on *L. monocytogenes*.

### 2.2. LIVE/DEAD Assay

Fluorescent staining can distinguish living, damaged and dead cells, which can indicate the viability of the bacteria [9]. The results depicted by the standard curves illustrated a good linear relationship (*R*^2^ = 0.99) between the green fluorescence intensity at 530 nm and the percentage of viable bacteria of *L. monocytogenes* ATCC 11915 and *L. monocytogenes* 001 (Appendix A). When exposed to 16, 32 and 64 μg/mL of sertraline, the green fluorescence intensity of *L. monocytogenes* 11915 decreased by 29.1%, 38.7% and 46.9%, respectively, and the green fluorescence intensity of *L. monocytogenes* 001 decreased by 45%, 47.9% and 50.9%, respectively (Figure 2).

### 2.3. Effect of Sertraline on the Bacterial Cell Membrane

ATP is essential for many cell functions including substance transportation across the cell membrane [10]. When the cell membrane is impaired, the intracellular ATP decreases as a result of the decreased synthesis and increased hydrolysis of the ATP and the loss of inorganic phosphate through the highly compromised permeable cell membrane [11]. Compared with the control group, the intracellular ATP concentrations (Figure 3) and the intracellular pH (pHin) (Figure 4) of *L. monocytogenes* were significantly reduced when the concentration of sertraline increased from 16 to 64 μg/mL. The pH_in_ of normal *L. monocytogenes* ATCC 11915 was 7.87. After exposure to 16, 32 and 64 μg/mL sertraline, the pH_in_ decreased to 7.55, 7.46 and 6.95, respectively. The pH_in_ of normal *L. monocytogenes* 001 was 7.13. After adding 16, 32 and 64 μg/mL sertraline, the pH_in_ decreased to 7.09, 6.91 and 6.82, respectively.

### 2.4. Biofilm Formation Testing

Compared with the control group, the percentage of biofilm formation of *L. monocytogenes* ATCC 11915 decreased by 67.9%, 50.4%, 12.8% and 9.4% after treatment with 64, 32, 16 and 8 μg/mL sertraline for 12 h, respectively; similarly, the biofilm formation percentage of *L. monocytogenes* 001 decreased by 86.0%, 50.9%, 23.3% and 5.2%, respectively (Figure 5).

### 2.5. FESEM Observation

Normal *L. monocytogenes* cells have a smooth surface, a complete structure and an elongated rod shape (Figure 6). After treatment with sertraline concentrations higher than 16 μg/mL, the cell surface of *L. monocytogenes* became irregular and showed varying degrees of contraction and intercellular aggregation. The percentage of damaged cells and the degree of damage increased as the concentration of sertraline increased. The microstructural observations were in accordance with the findings on the impaired cell membrane integrity, decreased intracellular ATP concentration and reduced pH_in_ under the application of sertraline as observed in this study.

### 2.6. Virulence Gene Expression

The results showed that sertraline had different effects on the 10 virulence genes (*hly*, *argA*, *prfA*, *degU*, *actA*, *flaA*, *sigB*, *ltrC*, *sufS* and *sufU*) of *L. monocytogenes.* Compared with the control group, sertraline at 0.1 and 1 μg/mL significantly inhibited the expression of these virulence genes in the *L. monocytogenes* ATCC 11915 strain (Figure 7). However, when the concentration was higher at 2–8 μg/mL, sertraline exposure weakened the inhibition effect on *actA*, *flaA*, *sigB*, *ltrC*, *sufS* and *sufU* gene expression. Especially when the sertraline concentration exceeded 4 μg/mL, the expression of *hly*, *argA*, *prfA* and *degU* genes was up-regulated.

Sertraline had slightly different inhibition effects on the virulence gene expression of *L. monocytogenes* 001 (Figure 7). Sertraline at 0.1 μg/mL resulted in down-regulation in the expression of *prfA*, *degU*, *actA*, *flaA*, *sigB*, *ltrC* and *sufS* genes. When the concentration was higher than 1 μg/mL, the inhibitory effect of sertraline on virulence gene expression was weakened. Sertraline at 4 and 8 μg/mL increased the expression of all evaluated virulence genes.

## 3. Discussion

Previous studies have reported on the antimicrobial activity of sertraline with a MIC value of 4–128 μg/mL for *S. aureus*, 8–128 μg/mL for *B. subtilis*, 128 μg/mL for *Candida albicans* and 4–256 μg/mL for *E. coli* [12,13]. The antimicrobial activity of sertraline is highly species dependent. In this study, the intrinsic antibacterial activity of sertraline against the *L. monocytogenes* strains was similar to the activity reported in *S. aureus* and *B. subtilis* [14]. In addition, consistent with previous studies, sertraline exhibited a bactericidal effect against *L. monocytogenes* at 2 × MIC concentrations. Although not completely elucidated, the intrinsic antimicrobial activity of sertraline may be due to the benzene rings in the structure [12].

The bacterial cell membrane keeps the internal environment of the cell stable and maintains the normal metabolic function and energy transfer of the cell [10,15]. When the cell membrane is damaged, phosphate bonds and ion gradients as well as the energy (such as pH and ATP) transfer will change [16]. Thus, intracellular ATP and pH_in_ are good indicators of the integrity of the cell membrane. In the present study, we observed that sertraline can interact with the cell membrane of *L. monocytogenes* as reflected by the decreased intracellular ATP and pH_in_. These findings are in accordance with previous studies, which demonstrated that sertraline can destroy the cell membrane of *H. pylori* [17] and, when combined with polymyxin, can significantly affect the ability of *Acinetobacter baumannii*, *K. pneumoniae* and *P. aeruginosa* to reshape their outer membranes [18]. In addition, sertraline is reported to cause *Candida* cell death by blocking the mitochondrial respiration and significantly decreasing transmembrane potential [19]. These results together indicate that sertraline most likely interacts with the bacterial cell membrane.

*L. monocytogenes* is a consistent source of cross-contamination, both in housing storage and food processing environments, and biofilm is easily formed on contaminated surfaces [5,20]. Previously, sertraline was shown to reduce the biofilm formation in different species, e.g., in *Candida* spp. [21]. In the present study, sertraline was found to have strong inhibitory effects on the biofilm formation ability of *L. monocytogenes* in food processing environments.

The virulence gene expression of *L. monocytogenes* involves several key steps: host cell adhesion and invasion, intracellular proliferation and motility and intercellular diffusion [22]. Specific bacterial factors are involved in each stage. For example, the *hly* gene, an important pathogenic factor, can help the bacteria escape the vacuole and interact with the host [23,24]. The *prfA* gene can regulate the expression of other virulent genes and control the biofilm formation [25]. The *actA* gene is required for the intracellular movement of *L. monocytogenes* in host cells [26]. The *agrA* gene affects biofilm formation and assists bacteria in intracellular invasion [1,27]. The *degU* and *flaA* genes are thought to stimulate the synthesis of bacterial flagella [28]. The *sigB* and *ltrC* genes have been shown to be involved in the low temperature adaptation of *L. monocytogenes* [29,30]. The *sufS* and *sufU* genes are associated with bacterial pathogenicity and virulence [31]. In this study, we found sertraline regulated the expression of the virulence genes of *L. monocytogene*; however, the regulation was not in a dose-dependent manner. In concentrations as low as 0.1 and 1 μg/mL, the expression levels of most virulence genes were down-regulated. However, high concentrations of sertraline increased the virulence gene expression. In a previous in vivo study, high concentrations of sertraline exacerbated pathological outcomes in chickens infected with resistant *E. coli* [30]. These results suggest sertraline might have multi-effects on virulence gene expression. Nonetheless, the inhibitory effect of a low concentration of sertraline on virulence gene expression indicates the potential application of sertraline on modulating the virulence of pathogens. Further modification of sertraline or synthesis of its structural analogue is expected to improve its inhibitory effect on pathogens.

## 4. Materials and Methods

### 4.1. Reagents and Bacteria

Sertraline (purity ≥ 98%) was purchased from Shanghai Macklin Biochemical Co., Ltd. (Shanghai, China).

The *L. monocytogenes* strains that were used are shown in Table 1. The strains were inoculated with brain heart infusion broth (BHI) and cultured at 37 °C for 16 h.

### 4.2. MIC and MBC

The MIC values of sertraline against the *L. monocytogenes* strains were determined with the broth microdilution method [32]. Briefly, the *L. monocytogenes* strains were grown aerobically overnight at 37 °C on BHI broth. Then, the colonies were suspended in 0.9% NaCl and adjusted to 0.5 McFarland standard with a Sensititre^TM^ nephelometer (Thermo-Fisher Scientific, Eugene, OR, USA). Subsequently, the suspensions were diluted 100-fold in Mueller–Hinton broth (MHB), and 100 μL of the dilution was transferred to the wells of a sterile 96-well plate that had different concentrations of sertraline in MHB (100 μL). The final concentrations ranged from 2 to 128 μg/mL. The positive controls contained bacteria inoculum only, whereas the negative controls contained MHB only. The lowest concentration of the compounds that resulted in no visible growth of the test organisms was determined as the MIC. The MBC was the lowest concentration at which microbial growth could not be observed on the medium [33]. All experiments were determined in biological triplicates.

### 4.3. Growth and Viability Assays

The growth and viability assays were performed as previously described [9]. Briefly, *L. monocytogenes* ATCC 11915 and a selected *L. monocytogenes* isolate (*L. monocytogenes* 001), previously obtained from a food product, were applied as test strains.

The assay was prepared at 37 °C for 12 h with continuous shaking. The strains were grown overnight, resuspended in BHI broth to reach an OD_600_ = 0.1 and then exposed to 0, 16, 32 and 64 μg/mL of sertraline. The samples (100 μL) were collected at seven different time points during the 12 h period. Then, the samples were serially diluted, spread on MHA plates and incubated at 37 °C for 18 h followed by successive counting. The experiment was performed in triplicate, and the results were expressed as the average Log_10_ CFU/mL.

### 4.4. Assessment of Membrane Integrity

The influence of the sertraline treatments on the membrane integrity of *L. monocytogenes* ATCC 11915 and *L. monocytogenes* 001 was assessed using the LIVE/DEAD *Bac*Light^TM^ Bacterial Vitality Kit (Thermo-Fisher Scientific, Eugene, OR, USA) as previously reported [34]. Briefly, standard samples were first prepared to construct a standard curve. The strain cultures were grown to the late exponential phase and then centrifuged, washed two times and re-suspended in 0.85% NaCl or 70% isopropyl alcohol (for the killed bacteria). Subsequently, the suspensions were incubated at room temperature for 1 h with mixing every 15 min. After incubation, the samples were pelleted two times with centrifugation (10,000× *g*, 10 min) and resuspended in NaCl to reach an OD_600_ = 0.5. Different viable cell proportions (0%, 10%, 50%, 90% and 100%) were utilized as the standard samples. A working stain solution (2×) was prepared by adding 6 μL of SYTO 9 and 6 μL of propidium iodide (PI) to 2 mL of filter-sterilized water.

The cultures of the strains were grown overnight and then adjusted to an OD_600_ = 0.5 followed by treatment with sertraline at 0, 16, 32 and 64 μg/mL for 15 min at 37 °C. After treatment, each culture was centrifuged and resuspended in 0.85% NaCl. Then, an equal volume of 100 μL of cell suspension and working stain solution (2×) was added to the black opaque 96-well microtitration plates (Corning, New York, NY, USA) and mixed thoroughly. The mixture was cultured in darkness at 25 °C for 15 min. The fluorescence was determined using a multifunctional enzyme marker (BioTek, Winooski, VT, USA). The green (530 nm) and red (630 nm) emission integral intensities of the suspension excited at 485 nm were obtained three times via wavelength measurement.

### 4.5. Measurement of Intracellular ATP Concentrations

The influence of sertraline on the intracellular ATP concentrations of *L. monocytogenes* ATCC 11915 and *L. monocytogenes* 001 was assessed as described previously [10]. Briefly, the overnight cultures of the strains were harvested with centrifugation (5000× *g*, 5 min). Then, the cells were washed three times with 0.1 mol/L of phosphate-buffered saline (PBS, pH 7.0) and resuspended in PBS to achieve an OD_600_ = 0.5. Subsequently, sertraline was added to each tube to achieve the final concentrations of 0, 16, 32 and 64 μg/mL and cultured at 37 °C for 30 min. The ATP was ultrasonically extracted on ice and centrifuged (5000× *g*, 5 min). The supernatant was kept on ice to avoid loss of ATP. The ATP content was determined using an adenosine triphosphate detection kit with a multifunctional enzyme marker (BioTek, Winooski, VT, USA) following the manual’s instructions (Beyotime Biotechnology, Shanghai, China). A good linearity was found between intracellular ATP content and luminescence (*R*^2^ = 0.99).

### 4.6. PH_in_ Measurements

The influence of sertraline on the pHin of *L. monocytogenes* ATCC 11915 and *L. monocytogenes* 001 was determined according to a modified method of Wang et al. [35]. First, the calibration curve was constructed by measuring a series of fluorescence intensities of the pH buffers with values in the range of 3 to 10. The buffers consisted of 50 mmol/L KCl, 50 mmol/L Na_2_HPO_4_·2H_2_O, 50 mmol/L glycine and 50 mmol/L citric acid, and they were adjusted with either NaOH or HCl. The pH_in_ and pH_out_ were equilibrated by adding 10 μmol/L valinomycin and 10 μmol/L nigericin.

A total of 300 μL of the overnight-cultured strains was transferred into 30 mL BHI broth and cultured at 37 °C for 8 h. After centrifugation (5000× *g*, 10 min), the cells were washed two times with 50 mmol/L HEPES buffer (containing 5 mmol/L EDTA, pH = 8) and resuspended in 20 mL buffer. Then, 3 μmol/L of the probe (carboxyfluorescein diacetate succinimidyl ester; cFDA-SE) (Meilunbio, Dalian, China) was added and cultured at 37 °C for 20 min. The cells were subsequently washed with 50 mmol/L potassium phosphate buffer added with 10 mmol/L MgCl_2_ (pH = 7.0), resuspended in 10 mL buffer and subsequently added with 10 mmol/L glucose and cultured at 37 °C for 30 min to eliminate the unbound cFDA-SE. The obtained particles were washed two times using the above mentioned method and suspended in 50 mmol/L potassium phosphate buffer on ice. Sertraline was added to the treated cell suspension to obtain the final concentrations of 0, 16, 32 and 64 μg/mL. Then, the mixture was transferred into a black opaque 96-well microtiter plate. After treatment for 20 min, the fluorescence intensity was measured under the excitation wavelengths of 440 and 490 nm with an emission wavelength of 520 nm at 25 °C by using a multifunctional enzyme marker (BioTek, Winooski, VT, USA). The pH_in_ was determined as the ratio of the fluorescence signals at the pH sensitive wavelength of 490 nm and pH insensitive wavelength of 440 nm. The fluorescence of the cell-free controls was measured and deducted from the fluorescence of the samples.

### 4.7. Biofilm Formation

The effect of sertraline on biofilm formation was investigated according to the method described previously [6,36]. Briefly, the concentration of the bacterial solution was adjusted to an OD_600_ = 0.5. Then, 100 μL of the bacterial solution was added into a 96-well plate and incubated at 37 °C for 6 h to form biofilms. Sertraline was added to the treated cell suspension to obtain the final concentrations of 0, 16, 32 and 64 μg/mL. MH broth and the bacterial solution were used as the negative and positive controls, respectively. After treatment for 16 h at 37 °C, the plate was washed 3 times with normal saline and dyed with 125 μL 1% crystal violet solution. The optical density at 595 nm was determined with a multifunctional microplate analyzer (BioTek, Winooski, VT, USA). The biofilm formation ability was the value after deducting background staining.

### 4.8. Field Emission Scanning Electron Microscope (FESEM) Analysis

The FESEM assays were prepared as previously reported with minor modifications [9]. Briefly, the cells (OD_600_ = 0.5) were treated with sertraline at 0, 16, 32 and 64 μg/mL and then cultured at 37 °C for 4 h. The cultured cells were centrifuged (5000× *g*, 10 min), washed two times with 0.85% NaCl, resuspended in 2.5% glutaraldehyde solution and cultured at −4°C for 10 h for fixation. Then, the cells were centrifuged (5000× *g*, 10 min) and dehydrated in gradient concentrations of ethanol (30%, 50%, 70%, 80%, 90% and 100%). Finally, the samples were fixed onto the FESEM support, sputter-coated with gold under vacuum and examined using a FESEM apparatus (Bruker, Berlin, Germany).

### 4.9. Virulence Gene Expression

The effect of sertraline at different concentrations on the expression of 10 virulence genes (*hly*, *argA*, *prfA*, *actA*, *degU*, *flaA*, *sigB*, *ltrC*, *sufS* and *sufU*) of *L. monocytogenes* was detected with real-time quantitative PCR using the primers listed in Appendix A [23].

First, the total RNA of *L. monocytogenes* cultures treated with sertraline at 0, 16, 32 and 64 μg/mL for 4 h was extracted with a commercial kit (Magen, GuangZhou, China) following the manufacturer’s instructions. The concentration and quality of the RNA were assessed with agarose gel electrophoresis and a Nanodrop 2000^®^ spectrophotometer (ThermoFisher Scientific, Waltham, MA, USA), respectively. Then, the cDNA was synthesized with a commercial kit (Vazyme Biotech Co., Ltd., Nanjing, China) following the manufacturer’s instructions.

The real-time PCR system was performed with the 2×Ultra SYBR Green qPCR Mix kit (CISTRO, GuangZhou, China) with the CFX96^TM^ Real-Time System (Bio-rad, Hercules, CA, USA). The *16S* rRNA gene was used as the reference gene [14]. The relative expression levels of the virulence genes were calculated according to the Ct values.

### 4.10. Statistical Analysis

All experiments were conducted in triplicate. The statistical analyses were performed in GraphPad Prism ver. 8.0.1 (San Diego, CA, USA). The data were presented as the mean ± standard deviation (n = 3), and the differences between the mean values were tested via the one-way ANOVA. The differences were considered to be significant at *p* < 0.05.

## 5. Conclusions

This study investigated the antimicrobial activity of sertraline against *L. monocytogenes.* Sertraline caused damage of the cell membrane and decreased the intracellular ATP and pH_in_ of *L. monocytogenes*. Moreover, sertraline significantly inhibited biofilm formation and regulated the virulence gene expression of *L. monocytogenes*. These results suggest that sertraline may potentially be used to control *L. monocytogenes* in food-processing environments, thereby reducing the risk of food contamination.

## Figures and Tables

**Figure 1 ijms-24-04678-f001:**
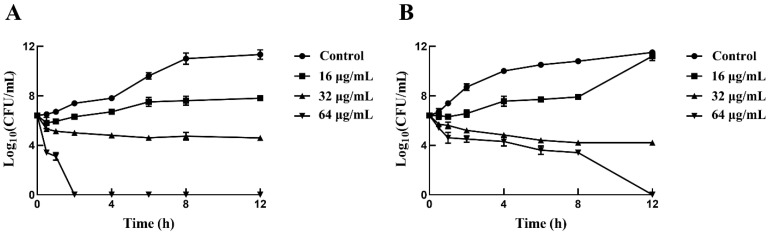
Time-killing curves for *L. monocytogenes* treated with sertraline in three concentrations (16 μg/mL, 32 μg/mL and 64 μg/mL) and untreated *L. monocytogenes*. (**A**) *L. monocytogenes* ATCC 11915 treated with sertraline; (**B**) *L. monocytogenes* 001 treated with sertraline. Values represent the average of three independent measurements.

**Figure 2 ijms-24-04678-f002:**
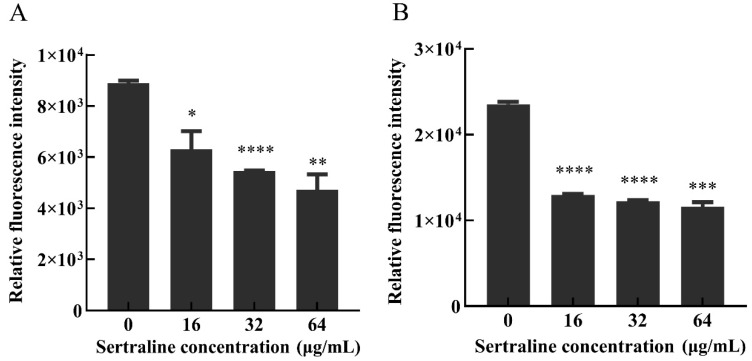
Effects of different concentrations of sertraline on *L. monocytogenes* cell membrane, which was indicated by the relative fluorescence intensity. (**A**) *L. monocytogenes* ATCC 11915 treated with sertraline; (**B**) *L. monocytogenes* 001 treated with sertraline. Each value is the mean of the three independent parallel control measurements, and the bars represent the standard deviation of the parallel samples, * *p* ≤ 0.05; ** *p* ≤ 0.01; *** *p* ≤ 0.001; **** *p* ≤ 0.0001.

**Figure 3 ijms-24-04678-f003:**
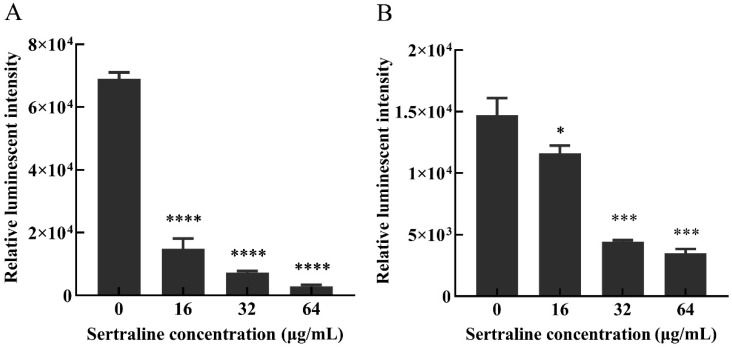
Effects of different concentrations of sertraline on the intracellular ATP content of *L. monocytogenes*, which was indicated by the relative luminescence intensity. (**A**) *L. monocytogenes* ATCC 11915 treated with sertraline; (**B**) *L. monocytogenes* 001 treated with sertraline. Each value is the mean of the three parallel control measurements, and the bars represent the standard deviation of the parallel samples, * *p* ≤ 0.05; *** *p* ≤ 0.001; **** *p* ≤ 0.0001.

**Figure 4 ijms-24-04678-f004:**
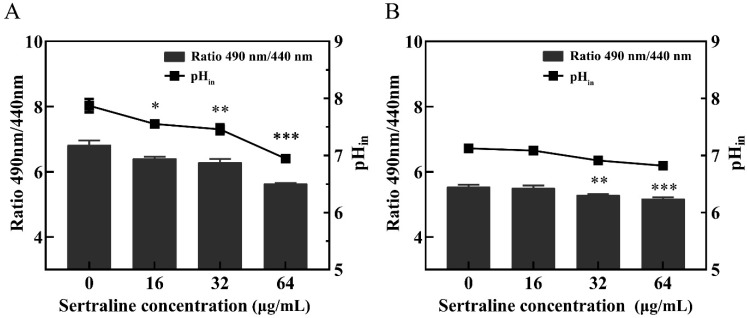
Effects of different concentrations of sertraline on pH_in_ of *L. monocytogenes*. The pH value was calculated by the ratio of the fluorescence emission intensity at 490 nm to 440 nm. In the Figure, the ratio of the fluorescence intensity is shown in the column, and the pH value in the cell is shown in the broken line. (**A**) *L. monocytogenes* ATCC 11915 treated with sertraline; (**B**) *L. monocytogenes* 001 treated with sertraline. Each value is the mean of the three parallel control measurements, and the bars represent the standard deviation of the parallel samples, * *p* ≤ 0.05; ** *p* ≤ 0.01; *** *p* ≤ 0.001.

**Figure 5 ijms-24-04678-f005:**
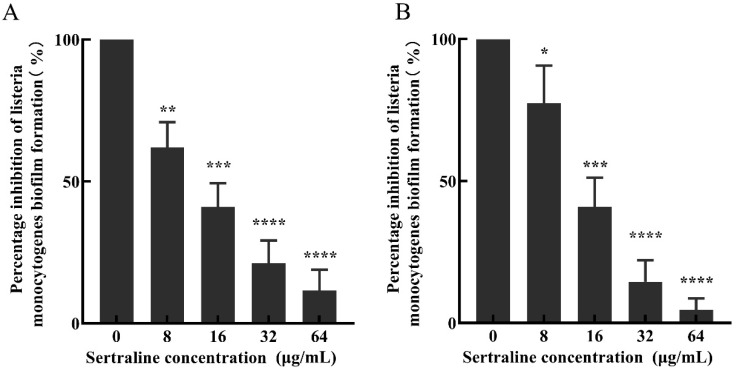
The inhibitory effect of sertraline at different concentrations on the biofilm formation of *L. monocytogenes* was expressed as the percentage of biofilm formation inhibition. (**A**) *L. monocytogenes* ATCC 11915 treated with sertraline; (**B**) *L. monocytogenes* 001 treated with sertraline. Each value is the mean of the three parallel control measurements, and the bars represent the standard deviation of the parallel samples, * *p* ≤ 0.05; ** *p* ≤ 0.01; *** *p* ≤ 0.001; **** *p* ≤ 0.0001.

**Figure 6 ijms-24-04678-f006:**
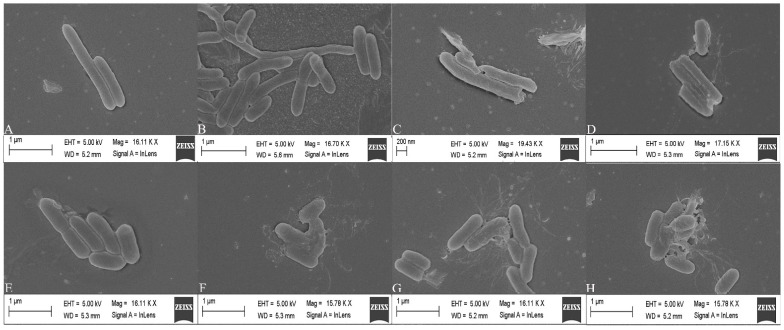
Scanning electron microscope images of *L. monocytogenes* cells treated with sertraline at different concentrations. (**A**) *L. monocytogenes* ATCC 11915 untreated; (**B**) *L. monocytogenes* ATCC 11915 treated with sertraline at 16 μg/mL; (**C**) *L. monocytogenes* ATCC 11915 treated with sertraline at 32 μg/mL; (**D**) *L. monocytogenes* ATCC 11915 treated with sertraline at 64 μg/mL; (**E**) *L. monocytogenes* 001 untreated; (**F**) *L. monocytogenes* 001 treated with sertraline at 16 μg/mL; (**G**) *L. monocytogenes* 001 treated with sertraline at 32 μg/mL; (**H**) *L. monocytogenes* 001 treated with sertraline at 64 μg/mL.

**Figure 7 ijms-24-04678-f007:**
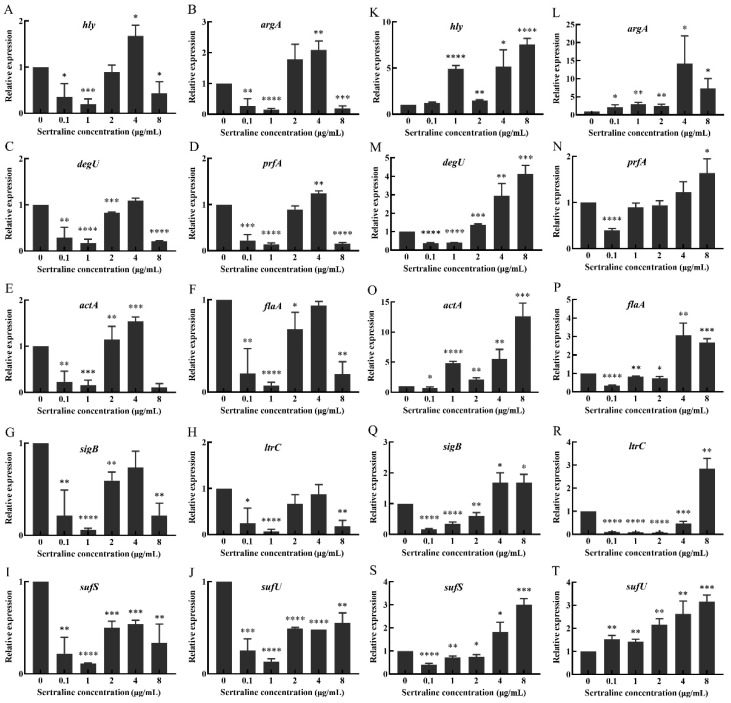
Changes in relative expression levels of virulence genes in *L. monocytogenes* treated with different concentrations of sertraline. (**A**–**J**) represents *L. monocytogenes* ATCC 11915, (**A**) *hly*; (**B**) *agrA*; (**C**) *degU*; (**D**) *prfA*; (**E**) *actA*; (**F**) *flaA*; (**G**) *sigB*; (**H**) *ltrC*; (**I**) *sufS*; (**J**) *sufU*. (**K**–**T**) represents *L. monocytogenes* 001, (**K**) *hly*; (**L**) *agrA*; (**M**) *degU*; (**N**) *prfA*; (**O**) *actA*; (**P**) *flaA*; (**Q**) *sigB*; (**R**) *ltrC*; (**S**) *sufS*; (**T**) *sufU*. Values represent the average of three independent measurements. * *p* ≤ 0.05; ** *p* ≤ 0.01; *** *p* ≤ 0.001; **** *p* ≤ 0.0001.

**Table 1 ijms-24-04678-t001:** MICs and MBCs of sertraline against different strains of *L. monocytogenes*.

Strain	Source	MIC (μg/mL)	MBC (μg/mL)
*L. monocytogenes* ATCC 11915	Standard strains	32	64
*L. monocytogenes* 001	Raw meat	32	64
*L. monocytogenes* 69	Frozen rice and noodle product	16	64
*L. monocytogenes* 60	Frozen rice and noodle product	16	64
*L. monocytogenes* 01	Frozen rice and noodle product	16	64
*L. monocytogenes* 82	Frozen rice and noodle product	32	64
*L. monocytogenes* 44	Frozen rice and noodle product	32	64
*L. monocytogenes* 46	Frozen rice and noodle product	16	64
*L. monocytogenes* EGDe	Frozen rice and noodle product	16	64
*L. monocytogenes* 10403s	Frozen rice and noodle product	16	64

## Data Availability

Data is contained within the article or Appendix A.

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
