# Peer review of "Antimicrobial Activity of Sertraline on Listeria monocytogenes"

_ijms, 2023, doi:10.3390/ijms24054678_

Round 1

Reviewer 1 Report

In this study, the authors investigated the effects of sertraline on planktonic cells, biofilms and virulence gene expression of Listeria monocytogenes.

Here are some general comments:

1. The article is difficult to understand and it is full of grammatical errors. The English usage is so poor. I strongly suggest improving the language usage.

2. The author only investigated the effects of sertraline on L. monocytogenes planktonic cells, which is not enough to reveal the antibacterial mechanism of sertraline.

3. In introduction section, the research background was not described well. The introduction was short of logic. This section should be rewritten.

4. Line 73: why select L. monocytogenes 001 for the next investigation?

5. why investigate the effects of sertraline on virulence gene expression?

6. In this paper, the effects of sertraline on planktonic cells, biofilms and virulence gene expression of L. monocytogenes. The three parts seem to be independent. I don’t understand why they appear in this article at the same time.

7. The article is full of grammatical and typographical errors, suggesting that the author lacks rigorous scientific attitude.

Specific comments

Title: The title seems like Chinglish. Please change a new title.

Abstract

Please modify the form of abstract. Words like “Objectives” “Result” should be deleted.

Line 14: please provide the full name of bacterial species name and “L. monocytogenesshould be italicized.

Line 24: please provide the full name of bacterial species name.

Line 24-26: meningitis, septicemia… are the clinical symptoms of listeriosis. Please rewrite this sentence.

Line 36-43: why list all strain names here? Make up the number of word?

Line 55: “Monocytogenes”?

Line 57: please provide the full name of MIC and MBC

Line 58: “…sertraline on L. monocytogenes…” The preposition “on” is not appropriate here.

Line 114: what is “Biotechnology,” ?

Line 116: what is PHin?

Line 161: maybe it is “2.9 virulence gene expression”

Line 183: please change a new subtitle.

Line 184: add a space between “strains” and “were”

Line 185-186: please rewrite this sentence.

Line 235: “Previousstudies” please check for format errors like this throughout. These format errors suggest the bad attitude to the paper of the author. Please check carefully.

Line 286: it is better not to use such statement.

Line 290: only sertraline was investigated in this study. The effects of sertraline structural analogue are unknow.

Line 298: “References 1 Adesokan…”

Table 1: in the column of “Source”, the origin of strain should be described, not “Strains were stored in the laboratory” or “Standard strains”

Author Response

Thank you for your valuable suggestions on the manuscript. For your suggestions, we have revised the manuscript and found a foreign counterpart to revise the language.

  1. The article is difficult to understand and it is full of grammatical errors. The English usage is so poor. I strongly suggest improving the language usage.

Response: We have invited a native English speaking professor help revise the language and we carefully checked the whole manuscript again and again to avoid errors.

  1. The author only investigated the effects of sertraline on L. monocytogenes planktonic cells, which is not enough to reveal the antibacterial mechanism of sertraline.

Response: We have changed the title as well as the whole manuscript to describe more precisely.

  1. In introduction section, the research background was not described well. The introduction was short of logic. This section should be rewritten.

Response: We have rewritten the introduction.

  1. Line 73: why select L. monocytogenes 001 for the next investigation?

Response: We select L. monocytogenes ATCC 11915 and a wild L. monocytogenes isolate 001 randomly for next investigation. Normally, only ATCC strains was needed. In this study, we aimed to see if sertraline has similar effect on wild strains and thus we randomly select a wild strain for comparison.

  1. why investigate the effects of sertraline on virulence gene expression?

Response: The effect of sertraline on virulence of strains has seldom been investigated. As L. monocytogenes is a foodborne pathogen, we aimed to illustrate it’s potential effect on virulence of L. monocytogenes. The results of this study firstly indicated low concentrations of sertraline at 0.1 μg/mL and 1 μg/mL significantly down-regulated the expression levels of virulence genes, prfA, actA, degU, flaA, sigB, ltrC, and sufS of L. monocytogenes.

  1. In this paper, the effects of sertraline on planktonic cells, biofilms and virulence gene expression of L. monocytogenes. The three parts seem to be independent. I don’t understand why they appear in this article at the same time.

Response: Sertraline has been reported to possess antimicrobial activity against a wide range of bacteria; however, it is not entirely clear how this compound performs its functions and little is known about their effects on biofilm and virulence genes expression. Biofilm and virulence as well as the growth of the L. monocytogenes strains are related. If sertraline can also inhibit biofilms and virulence gene expression, it might be a promising approach to combat L. monocytogenes.

  1. The article is full of grammatical and typographical errors, suggesting that the author lacks rigorous scientific attitude.

Response: We have invited a native English speak professor help the grammar and language problems and we believe it now meet the requirement of this Journal.

Specific comments

Title: The title seems like Chinglish. Please change a new title.

Response: We have modified the title.

Abstract

Please modify the form of abstract. Words like “Objectives” “Result” should be deleted.

Response: We have revised this sentence.

Line 14: please provide the full name of bacterial species name and “L. monocytogenes“ should be italicized.

Response: We have revised this sentence.

Line 24: please provide the full name of bacterial species name.

Response: We have revised this sentence.

Line 24-26: meningitis, septicemia… are the clinical symptoms of listeriosis. Please rewrite this sentence.

Response: We have modified the sentence.

Line 36-43: why list all strain names here? Make up the number of word?

Response: We have modified the sentence.

Line 55: “Monocytogenes”?

Response: We have revised this sentence.

Line 57: please provide the full name of MIC and MBC.

Response: We have provided the full name of MIC and MBC here.

Line 58: “…sertraline on L. monocytogenes…” The preposition “on” is not appropriate here.

Response: We have revised this sentence.

Line 114: what is “Biotechnology,” ?

Response: We have revised this sentence.

Line 116: what is PHin?

Response: We have added explanation.

Line 161: maybe it is “2.9 virulence gene expression”

Response: We have revised this sentence.

Line 183: please change a new subtitle.

Response: We have modified the subtitle here.

Line 184: add a space between “strains” and “were”

Response: Revised accordingly..

Line 185-186: please rewrite this sentence.

Response: We have modified this sentence.

Line 235: “Previousstudies” please check for format errors like this throughout. These format errors suggest the bad attitude to the paper of the author. Please check carefully.

Response: We have carefully revised whole manuscript to avoid format errors.

Line 286: it is better not to use such statement.

Response: We have modified this sentence.

Line 290: only sertraline was investigated in this study. The effects of sertraline structural analogue are unknow.

Response: We have modified this sentence.

Line 298: “References 1 Adesokan…”

Response: We have modified the text here.

Table 1: in the column of “Source”, the origin of strain should be described, not “Strains were stored in the laboratory” or “Standard strains”

Response: We have modified Table 1.

Reviewer 2 Report

This work is devoted to the study of the antibacterial activity of different doses of sertraline in relation to the bacteria of Listeria monocytogenes.  In addition, they analyzed the effect on biofilm formation and virulence gene expression of L. monocytogenes strains.

There are similar works in the literature for other bacteria.

-Lines 5-12, last names have letters, and then followed by numbers

- The design of the work is not convenient for reading (figures and a table are located at the end of the manuscript)

- In the abstract, it is worth removing the words “Objectives” and “Conclusion” themselves

- Keywords are missing

- Introduction is very short

- No ATP decryption

- Perhaps it is worth explaining the concept of "biofilm"

- Incorrect links in the text

- Discuss the effectiveness as well as the advantages and disadvantages of the method compared to other methods

-p.161-?

Author Response

Thank you for your valuable suggestions on the manuscript. We have revised the manuscript in response to your suggestions.

Lines 5-12, last names have letters, and then followed by numbers

Response: We have modified these texts.

- The design of the work is not convenient for reading (figures and a table are located at the end of the manuscript)

Response:The figures and a table are acceptable by this journal to be located at the end of the manuscript.

- In the abstract, it is worth removing the words “Objectives” and “Conclusion” themselves

Response:We have removed “Objectives” and “Conclusion”.

- Keywords are missing

Response:We have added keywords.

- Introduction is very short

Response:We have added introduction.

- No ATP decryption

Response:We have added ATP description in methods and results.

- Perhaps it is worth explaining the concept of "biofilm"

Response:We have added the concept of "biofilm".

- Incorrect links in the text

Response:We have modified the whole manuscript carefully.

- Discuss the effectiveness as well as the advantages and disadvantages of the method compared to other methods

Response:The methods we used were general methods that have been widely used in other studies (as listed in references). As these methods has not been used to analyze the effect of sertraline on Listeria, we didn’t focused on compareration of the methods.

-p.161-?

Response:We have modified this subtitle.

Round 2

Reviewer 1 Report

The authors have revised the manuscript carefully.